# Risk Management of Methane Reduction Clean Development Mechanism Projects in Rice Paddy Fields

Eun-Kyung Jang [1,*] , Emily Marie Lim [2] , Jumi Kim [3] , Moon-Jung Kang [4], Gayoung Choi [4] and Jooyeon Moon [4]

1   Wood Industry Division, National Institute of Forest Science, Seoul 02455, Republic of Korea
2   Graduate School of Environmental Studies, Seoul National University, Seoul 08826, Republic of Korea; emilymarielim@gmail.com
3   Department of Forestry and Landscape Architecture, Konkuk University, Seoul 05029, Republic of Korea; kimzoom9@gmail.com
4   Global Business Center, National Institute of Green Technology, Seoul 07328, Republic of Korea; kangmj@nigt.re.kr (M.-J.K.); choigayoung@nigt.re.kr (G.C.); jooyn@nigt.re.kr (J.M.)
*   Correspondence: mentaka1209@gmail.com; Tel.: +82-10-3838-8547

**Abstract:** Agriculture accounts for the largest share of anthropogenic methane emissions. Rice paddy fields emit a significant amount of methane gas worldwide. Changing paddy water management practices has an enormous potential to reduce greenhouse gases. The clean development mechanism (CDM) project uses a market mechanism to reduce methane through private participation. There are various risks associated with private investment in CDM projects, although carbon credits as an economic incentive assist in mitigating some of these risks. Farmer participation plays a key role in the success of paddy water management projects in rural areas; however, despite the significant potential to reduce global methane emissions, very few projects have been implemented. When designing a Sustainable Development Mechanism (SDM) system, it is crucial to understand why the market mechanism in the existing CDM projects has failed. This study identifies and categorizes the risks and barriers to paddy water management in CDM projects and analyzes risk management options in CDM projects in India, Indonesia, and Mozambique. The results of this study showed that aside from economic risks, barriers to the application of technology in the field pose critical risks. The lack of knowledge and implementation experiences in rural areas increases barriers to practice. This in turn causes risk of difficulties in technology transfer which can be alleviated by improving awareness and introducing new knowledge through education and training in rural project implementation. Additionally, we highlight the importance of international efforts to build governance between the private and public sectors and promote technology transfers through multi-stakeholder engagement. This study provides specific information to encourage methane reduction worldwide and vitalize rice paddy water management in carbon reduction projects.

**Keywords:** clean development mechanism (CDM); global methane pledge; alternate wetting and drying (AWD); risk management; water management governance; climate change mitigation; adaptation

## 1. Introduction

Sharp and rapid reductions in methane emissions are essential to limit global warming by 1.5 °C. While carbon dioxide ($CO_2$) has a long-lasting effect, methane ($CH_4$) possesses 80 times the warming power of $CO_2$ within the first 20 years of entering the atmosphere [1]. Methane is setting the pace for near-term global warming [2].

Methane has contributed to a 30% rise in global temperatures since the Industrial Revolution; although methane has a shorter atmospheric lifetime, it absorbs more energy than $CO_2$ when in the atmosphere. The concentration of methane in the atmosphere is currently 2.5 times greater than that in the pre-industrial times. The concentration has been increasing steadily over the years and was expected to break all records in 2021 [3]. The Sixth Assessment Report from the Intergovernmental Panel on Climate Change (IPCC)

revealed that based on 2021 data, the atmospheric methane concentration (1896 ppb) was higher than that of both $CO_2$ (415 ppb) and nitrous oxide (335 ppb). The report emphasized that achieving net-zero emission targets necessitates significant reductions not only in $CO_2$, but also methane and other greenhouse gas emissions [4]. The most recent assessment estimates annual global methane emissions at approximately 580 Mt [5]. Methane contributes to approximately 16% of the radiative emissions from greenhouse gases [3]. As the residence time of methane in the atmosphere (around 12 years) is shorter than that of other greenhouse gases, such as $CO_2$, reducing methane emissions can make a significant contribution to curbing global warming [5].

The Global Methane Pledge was launched in November 2021 at the United Nations (UN) Climate Change Conference in Glasgow (COP26) to reduce methane gas emissions. This pledge was led by the UN and the European Union, and now has over 110 country participants who are collectively responsible for 45% of methane emissions [5]. By signing this pledge, countries commited to working together to collectively reduce anthropogenic methane emissions by at least 30% below 2020 levels by 2030.

Global methane emissions are increasing, especially from anthropogenic sources in the agricultural sector, and waste and biomass industries. Around 60% of emissions originate from anthropogenic sources such as landfills, biomass, rice agriculture, and fossil fuel use, and around 40% originate from natural sources [3]. Anthropogenic methane emissions by sector, as reported to the UN Framework Convention on Climate Change (UNFCCC), show that anthropogenic emissions from the agriculture sector are the highest (Figure 1) [6]. The Global Methane Assessment reports that the agricultural sector constitutes about 40% of global anthropogenic emissions of methane, with rice cultivation accounting for 8% [7].

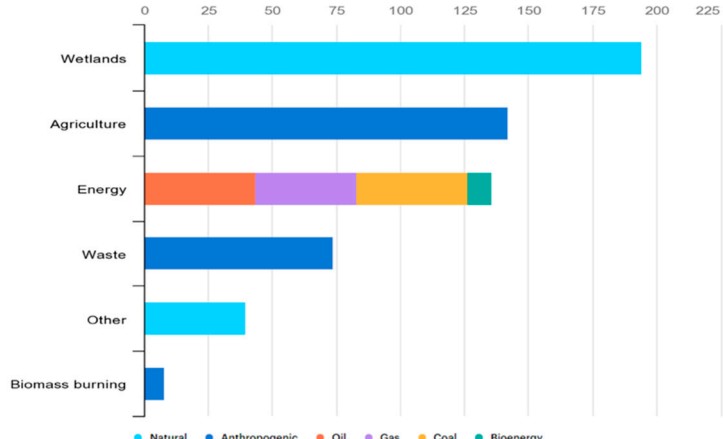

**Figure 1.** Sources of methane emissions in 2021 (unit: Mt $CH_4$) (source: IEA Global Methane Tracker, 2022).

Flooded rice paddy fields account for 12% of global anthropogenic methane emissions [7]. The methodology for securing carbon credits is registered in the UNFCCC Clean Development Mechanism (CDM) as AMS-III.AU: Methane emission reduction by adjusted water management practice in rice cultivation. Despite the agricultural sector accounting for the largest proportion of anthropogenic emissions of methane, most of the methane reduction projects through the CDM originate from the energy and waste industries, with less than 10% of projects originating from the agriculture sector. Water management projects in rice paddies are the largest source of methane in developing countries where rice is the main agricultural crop. Although rice farming is responsible for a significant proportion of methane emissions at a scale comparable to each of the energy sub-sectors of gas, oil, and coal [7], CDM projects for methane reduction have not been implemented owing to implementation difficulties and scarce funding. Therefore, it is crucial that the potential contributions to CDM methane reduction projects on rice paddy water management are not overlooked. To date, three rice paddy water management projects have been submitted

to the CDM, but only for three countries: India, Indonesia, and Mozambique. Each country presents unique characteristics and challenges in rice cultivation, making them relevant for studying the risks involved.

India is one of the largest contributors to the global methane budget; India's methane emissions in 2019 totaled 658.09 $MtCO_2e$, ranking it as the fourth largest methane emitter after China, the United States, and Russia. In 2016, the agricultural sector in India was responsible for approximately 14.3% of emissions in $CO_2$ equivalent, where rice cultivation was identified as the tenth key emission category within the country. India boasts the largest area dedicated to rice cultivation globally [8]—a total of 43.19 million hectares. In 2016, this subcategory contributed to 3% of the country's total emissions (71.32 $MtCO_2e$) and 17.5% of greenhouse gas emissions within the agricultural sector [9]. India plays a crucial role in addressing methane emissions globally, as it accounts for 26% of global rice production [10].

In Indonesia, rice cultivation covers an area of 13.84 million hectares [8]. Methane emissions from rice cultivation account for 0.99% (25.24 $MtCO_2e$) of the country's total emissions and 23.96% of emissions are from the agriculture sector [11]. Between 2014 and 2018, the harvested rice paddy area experienced an annual growth rate of 3.67%, while production increased by 2.26% [11]. The expansion and contraction of paddy field land area occurred in distinct periods, resulting in a net decrease of 628,959 hectares during the 2015–2019 period. These fluctuations in the area of cultivation could have significant implications for food security.

Mozambique, despite not being typically highlighted in discussions of major methane emitters or rice producers, presents unique circumstances that justify its inclusion as a case study country. Rice is one of the major food crops cultivated in Mozambique—rice cultivation area occupies 290,000 hectares and has increased by approximately 5% between 2019 and 2023 [10]. As Africa's population is predicted to double by 2050 [12], food security concerns assume critical importance in the region. Population expansion and the increase in middle-class consumers in Mozambique has contributed to significant rice deficits and a heavy reliance on rice imports [13,14]. Analyzing the risks associated with rice paddy CDM projects in Mozambique can provide valuable insights into managing these emissions in a different regional context.

Addressing methane emissions arising from rice cultivation can significantly influence overall methane abatement strategies, not only in the case countries but also in other regions where rice is a prominent crop. This study explores the management of risks associated with rice paddy water management in CDM projects by identifying the obstacles to paddy water management in CDM projects and analyzing management options that can lower risks in these projects in India, Indonesia, and Mozambique.

## 2. Theoretical Background

### 2.1. Alternate Wetting and Drying (AWD)

There are several validated management options to mitigate methane emissions in rice cultivation, which maintain or improve yields, enhance profitability, and increase climate resilience. For example, by integrating a locally adapted water-saving technology for rice production, such as alternate wetting and drying (AWD), methane emissions can be reduced by 30–70%. Most methane emissions are caused by biological factors, such as the activities of methanogenic bacteria. The mechanism of anaerobic oxidation of methane (AOM) involves a physical association between anaerobic methanotrophic archaea and sulfate-reducing bacteria [15]. Generally, methanogens produce methane by decomposing organic matter in anaerobic (oxygen-poor) environments [15] such as flooded rice paddy fields (see Figure 2). Approximately 8% (30 Tg) of anthropogenic methane emissions are generated by paddy fields [16]. Rice paddy water management technology that reduces methane generation in paddy fields involves temporarily sifting or filtering paddy water when little water is needed and at a level that does not affect yield. When the ground dries out owing to the lack of water, oxygen in the air diffuses into the ground and reduces

methane emission; the longer the drying period, the greater this effect. AWD lowers methane emissions through the periodic introduction of aerobic soil conditions.

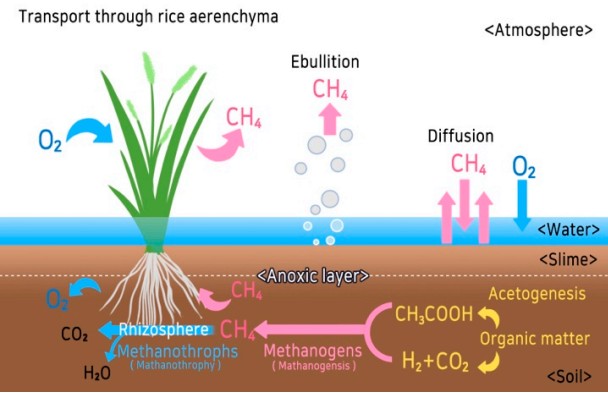

**Figure 2.** The methane generation mechanism in a paddy field (source: Original figure).

In AWD, also known as controlled or intermittent irrigation, the rice field is drained naturally, and non-flooded soil can be maintained from 1 to more than 10 days. Farmers monitor water depth using a perforated field water tube and re-flood the field up to a depth of around 5 cm at the time of flowering, when the water level is 15 cm below the soil surface [17]. This cycle is repeated throughout the cropping season except 1 week before and after flowering. Maintaining the 15cm water level below soil surface threshold is known as "safe AWD" and does not lead to yield declines if conducted correctly [18,19]. Aside from significant methane reductions, AWD is a means of adapting to water scarcity that brings about economic benefits. Compared to cultivating rice through the continuous flooding of paddy water, AWD practices can reduce methane emissions by 66–72% [20] and reduce the volume of irrigated water by 20–50% with minimal yield losses [18]. In Vietnam, AWD has assisted in improving farm profitability by up to 13% (around USD 100 per ha). AWD practices in the Philippines resulted in increased economic yields owing to irrigation costs, especially for pump users [21].

### 2.2. Risk Management

Risk can be defined in several ways; it may be defined as the potential for negative consequences resulting from a reaction to an event [22]; the probability that a substance or situation will cause harm under certain conditions [23]; the likelihood that an event will affect objectives [24]; or a combination of exposure and hazard [25]. Risk is perceived as uncertainty related to the possibility of harmful consequences.

Implicit to the definitions of risk is an understanding that risk is context-dependent [26]. As CDM projects for water management in rice paddy fields are uncommon, there is a dual risk associated with both the operation of innovative water management projects in rural areas and the implementation of CDM projects. The risk of innovative project in a rural area is defined as an uncertain event or condition that affects one or more project objectives, such as the scope of occurrence, schedule, costs, and quality [27]. Previous studies have classified the risks associated with the development and management of water management projects in rural areas into financial, political and institutional, technical, environmental, and social components [28]. These risks are summarized in Table 1.

**Table 1.** Risks in agriculture projects.

| Category | Risk for Agriculture Projects |
|---|---|
| Financial | Liquidity problem [29,30]<br>Economic disadvantage [31]<br>Cost overruns [21]<br>Inflation, and interest rates [32]<br>protectionism [33]<br>Counter-party risk [34]<br>High energy and water costs [35]<br>Late payments [35]<br>Crop loss [35,36]<br>Limited funding [37,38] |
| Political and institutional | Political instability [39]<br>Corruption [40]<br>Agriculture policies [41]<br>Changes in policies and regulations [33,42]<br>Governance conflicts and/or conflicts of interest [43,44]<br>Lack of farmer incentives [21] |
| Technical | Poor machine operation [35]<br>Contractor performance [35]<br>Poor construction methods [45]<br>Poor communication and coordination [32]<br>Material shortages [32]<br>Supply chain breakdown and limited accessibility [46] |
| Environmental | Climate change [47]<br>Adverse weather conditions [32,48–51]<br>Natural risks, such as droughts, floods, cyclones, and storms [48,52]<br>Soil quality [36] (Bebbington et al., 2006)<br>Degradation and loss of habitats and landscapes [35,53] |
| Social | Poverty or social exclusion [54]<br>Rural depopulation and aging [55]<br>Lack of knowledge and experience [34,56]<br>Loss of access to property [26]<br>Conflict between cultures [43]<br>Internal armed conflict [35,57]<br>Limited access to information and communication [57]<br>Displacement and resettlement [58]<br>Protest actions [59]<br>Violation of human rights [60]<br>Theft [35] |

Few studies have examined the risks faced by CDM projects. Some studies have categorized risk into three stages: planning, implementation, and CDM administration (CDM registration, monitoring, and verification). Risks at the planning stage include baseline [61] and technology transfer risk [62]. Planning stage risk refers to methodological applicability, the selection of the target site, the application of the methodology, and financial aspects [63]. At the project implementation stage, studies have identified project risk [64], country and policy risk [61,63,65,66], and sustainability risk in the social and environmental sector [63,67]. The registration, monitoring, and verification processes at the CDM administration stage are summarized in Table 2.

**Table 2.** Clean development mechanism (CDM) risk.

| | | **Types of Risk** | **Citations** |
|---|---|---|---|
| Planning | Technology/Methodology | Technology transfer risk | [62] |
| | | Baseline risk | [61] |
| | Finance | Financial risk | [63] |
| | | Investment risk | [62,66] |
| Implementation | Projects | Project risk | [64] |
| | | Country risk/Policy risk | [61,63,65,66] |
| | | Socioeconomic risk | [63] |
| | | Sustainability risk | [67] |
| CDM administration | CDM registration, Monitoring, and Verification/ Certification | Crediting risk | [68] |
| | | Certified emission reductions (CER) risk/Business risk | [61,63,67] |
| | | Certification risk/ CDM registration risk | [61,67–69] |

*2.3. Scope and Applicability of Adjusted Water Management under CDM*

Introduced by the Kyoto Protocol in 1997, the CDM is the only project-based mechanism that involves non-Annex 1 parties (or developing countries) without any Greenhouse Gas (GHG) emission obligations to fulfill. It allows Annex 1 (or developing countries) to reap mitigation benefits from a CDM project implemented in a developing country through the purchase of certified emission reduction (CER) credits to meet their emission reduction targets or emission caps. While promoting sustainable development and reduction in GHG emissions, the CDM provides industrialized countries the flexibility to achieve their emission reduction targets. CDM projects undergo rigorous examination [70]. Submitted projects must follow an existing methodology listed under the CDM (new methodologies may also be submitted for approval). Among the methodologies listed under the CDM, methane emission reduction using the adjusted water management practice in rice field is listed as a small-scale methodology.

For a methodology to be applicable, three scopes of project activities and seven essential conditions must be satisfied. The three scopes of project activities are (a) rice farms that change their water regime during the cultivation period from continuously to intermittent flooding and/or a shortened period of flooded conditions; (b) the AWD method and aerobic rice cultivation methods; and (c) rice farms that change their rice cultivation practice from transplanted seedlings to directly seeded rice (DSR) [71].

To satisfy the requirements, the projects also need to satisfy conditions related to the target site, project practice activities, and accounting method and reduction amount, as shown in Table 3.

**Table 3.** Requirements of projects for reducing methane in paddy fields.

| Category | Applicability |
| --- | --- |
| Site requirements | 1. Rice cultivation in the project area is predominantly characterized by irrigated, flooded fields for extended periods during the growing season, i.e., farms whose water regimes can be classified as upland or rain-fed and deep water are not eligible for applying this methodology based on a representative survey conducted in the geographical region of the proposed project or by using national data. The project area characterization also needs to include information on the pre-season water regime and organic amendments applied so that all the dynamic parameters shown in Table 2 are covered by the baseline study. |
| | 2. The project's rice fields are equipped with controlled irrigation and drainage infrastructure so that appropriate dry/flooded conditions can be established on the fields in both dry and wet seasons. |
| Conditions of practice activities | 3. The project activity does not lead to a decrease in rice yield, nor does it require the farm to switch to a cultivar that has not been grown before. |
| | 4. Training and technical support provided during the cropping season that delivers appropriate knowledge in field preparation, irrigation, drainage, and use of fertilizer should be available to the farmer as part of the project activity. It should be documented in a verifiable manner (e.g., protocol of training, documentation of on-site visits). In particular, the project proponent must be able to ensure that the farmer themselves, or with the help of experienced assistance, is able to determine the crop's supplemental fertilization needs (e.g., nitrogen). The applied method must assess the fields' fertilizer needs using, for example, a leaf color chart, photo sensor, or testing stripes. Alternatively, a procedure to ensure efficient fertilization is selected that considers the specific cultivation conditions in the project area backed by scientific literature or official recommendations. |
| | 5. Project proponents should provide assurance that the introduced cultivation practices including the specific cultivation elements, technologies, and use of crop protection products are not subject to any local regulatory restrictions. |
| Conditions of accounting method and reduction amount | 6. Except in cases where the default value approach indicated in Section 6.1.2 in [4]"Emission reductions using IPCC tier 1 approach or default values" of the AMS-III AU is chosen for emission reduction calculations, project proponents have access to infrastructure to measure $CH_4$ emissions from reference fields using closed-chamber method and laboratory analysis. |
| | 7. Aggregated annual emission reductions of all fields included under one project activity should be less than or equal to $60\,kt\,CO_2$ Eq. |

(Source: Revised from the CDM methodology of AMS-III AU [71].)

## 3. Methods

In this study, the risks associated with the management of rice paddy CDM projects were identified based on the risk management method. The process steps for the risk management of rice paddy fields in CDM projects were divided into project planning, project implementation, project reporting, and evaluation. Detailed management components were identified and cases of risk management were investigated.

Among the management components for each risk management stage, project feasibility was set as a core criterion in the project planning stage. Project feasibility determines whether a project is eligible. Project validity is determined by examining whether a methodology that can be applied to the CDM project has been prepared, and whether the water management project planned by the operator can be applied to this methodology. In rice

paddy water management projects, the amount of methane to be reduced occurs in the rice fields. The AMS-III AU, the only available technical methodology under the CDM, identifies three cultivation methods to avoid methane production in paddy fields: (1) changing the cultivation practice from a transplanted seedling method to a direct sowing method, (2) temporarily draining water from the paddy field to prevent anaerobic bacteria from producing methane, and (3) controlling the amount of nitrogenous fertilizer used.

In this study, only the activities related to the temporary draining of water from the rice fields were targeted. For the case analysis, the study targeted three projects that were applied to the CDM and used the AMS-III AU methodology.

The analytical framework of this study is divided into project planning, implementation, reporting and evaluation stages. When designing a CDM project, the biggest risk in the project planning stage lies in whether an applicable methodology has been established. Once applicable methodologies are identified, the methodological applicability of the proposed project needs to be evaluated. This can be assessed according to the essential conditions for methane reduction in rice cultivation projects (see Table 3), which represents the main risks associated with CDM project registration.

In determining methodological applicability of projects to reduce methane in rice paddy fields, the following criteria are analyzed: baseline conditions, irrigation and drainage infrastructure, no impact on rice production, farmer education, compliance with local regulations, securing coefficients (emission factor), and small-scale CDM requirements. At the same time, additionality should be evaluated. In CDM, additionality is evaluated based on technical, practical, economic, and legal/institutional aspects. In this context, risks in the planning stage can be broadly classified into those associated with the applicable methodology, methodological applicability, and project additionality. The additionality analysis refers to the criteria presented in paragraph 10 of TOOL21: Demonstration of additionality of small-scale project activities, version 13.1. It states that "project participants shall provide an explanation that the project activity would not have occurred anyway due to at least one of the following barriers: (a) Investment barrier: a financially more viable alternative to the project activity would have led to higher emissions; (b) Technological barrier: a less technologically advanced alternative to the project activity involves lower risks due to the performance uncertainty or low market share of the new technology adopted for the project activity and so would have led to higher emissions; (c) Barrier due to prevailing practice: prevailing practice or existing regulatory or policy requirements would have led to implementation of a technology with higher emissions; (d) Other barriers: without the project activity, for another specific reason identified by the project participant, such as institutional barriers or limited information, managerial resources, organizational capacity, financial resources, or capacity to absorb new technologies, emissions would have been higher".

In the project implementation stage, risk management methods in the financial, political, institutional, technical, environmental, and social sectors were evaluated in the context of risks associated with rural projects. In this study, we focused on financial and technical risks to evaluate CDM risks, as well as social risks in a rural context. In the project reporting and evaluation stage, we examined monitoring, reporting, and verification (MRV), certification, and the possibility of utilizing carbon credits, which were mainly covered in previous studies. We also focused on risk management measures in the MRV sector, which are important for credit issuance, as we found no monitoring reports and carbon credits issued for paddy management CDM projects. This research considered the risks in the steps of the implementation and reporting and evaluation based on the contents designed in the project planning step. Table 4 summarizes the analysis framework. This study only analyzed the risk of components in project planning, which may include the risk components of project implementing and project reporting and evaluating: applicable methodology, methodological applicability, and project additionality.

**Table 4.** Analysis framework for risk management of adjusted water management in paddy fields.

| Category | Components |
|---|---|
| Project planning | Applicable methodology<br>Methodological applicability<br>Project additionality |
| Project implementing | Financial readiness<br>Political and institutional readiness<br>Technical readiness<br>Environmental readiness<br>Social readiness |
| Project reporting and evaluating | Monitoring, reporting, and verification (MRV)<br>Certification<br>Possibility of utilizing carbon credits |

## 4. Results

### 4.1. Applicable Methodology

All CDM cases for methane reduction in rice cultivation use AMS-III AU, the only registered CDM methodology.

### 4.2. Methodological Applicability

The target of AWD as a CDM project is based on the baseline of paddy fields under flooded conditions or improved water management when compared against the existing adjusted water management methods. For example, if the baseline is a flooded paddy field, the project activity involves draining water from the paddy field more than once. If one draining cycle has been completed at the baseline assessment, indicating an improved water management practice, additional methane reductions can be observed when more than two subsequent draining cycles are conducted during the project's implementation. The study examined cases in India, Indonesia, and Mozambique, where the baseline was flooded paddy fields.

The paddy fields in the target site must be equipped with irrigation and drainage infrastructure. The irrigation and drainage facilities in the case study of India were satisfactory. In Indonesia, rice fields with irrigation infrastructure were targeted. In Mozambique, the irrigation and drainage facilities were in the planning stage, and would be installed once the project is implemented in the future.

As for the production of rice, measures were taken not to affect rice production by harvesting under the same conditions as the baseline (cases in India and Indonesia). Farmers were provided with training and education on newly applied technologies (cases in India and Indonesia). Project activities were designed in compliance with local laws and regulations (cases in India and Indonesia). To calculate the carbon potential, the Indonesia case provided the emission factor of baseline and project based on preliminary projects, and the India case provided a chamber method and a laboratory analysis to obtain the $CH_4$ emission values from reference fields. The case study in India used the IPCC's default coefficient to calculate the carbon potential at the planning stage. Mozambique only formulated a plan to carry out the project according to the criteria for rice production, farmer education, compliance with local regulations, and coefficient development (specific methods were provided). All cases satisfied the carbon reduction requirements for small-scale CDM projects (India, Indonesia, and Mozambique). The results are summarized in Table 5.

**Table 5.** Methodological applicability by CDM projects in adjusted water management.

| Category | | India | Indonesia | Mozambique |
|---|---|---|---|---|
| Site requirements | Baseline | Continuously flooded paddy fields | Continuously flooded paddy fields Changed to moist conditions, intermittent irrigation, and flooding combined | Continuously flooded paddy fields |
| | Facilities for irrigation and drainage | Irrigation and drainage facilities satisfactory | Irrigation system satisfactory | Component Project Activities (CPA) will be equipped with an irrigation and drainage facility |
| Practice activities | Impact on rice production | No impact on rice production The cultivation method used in the CPA involved in the PoA does not lead to a decrease in rice yield. The rice cultivar /variety was not impacted by the project cultivation method | Same as the baseline of harvest methods and no impact on production amount Do not change the rice varieties and do not deal with modified varieties Same as the baseline, that is, locally available, traditional or new breed rice varieties | CPA will comply with the condition of no impact on rice production |
| | Farmer education | Provide training and technical support | Provide training and education for farmers who participated | Training will be provided |
| | Compliance with local regulation | Not in conflict with any laws or regulations in India | Neither the project activity as a whole nor its elements conflict with local laws or regulations | Agriculture practices will comply with local regulations |
| Accounting method and reduction potential | Coefficient development | Using chamber method and laboratory analysis to obtain the $CH_4$ emissions from reference fields | Providing emission factor of baseline and project based on preliminary projects | Applicability condition will be followed for the CPAs |
| | Small-scale CDM emission standards | Satisfied according to SSCDM (less than 60 kt $CO_2$ Eq) | Satisfied according to SSCDM (less than 60 kt $CO_2$ Eq) | Expected to satisfy SSCDM requirement (less than 60 kt $CO_2$ Eq) |

*4.3. Project Additionality*

Additionality in CDM projects was divided into barriers to investment, technology, practical implementation, and others. In the case of rice water management projects, additionality was demonstrated in all three aforementioned areas. Particularly, the high risks associated with field projects in rural areas constitute barriers associated with the uptake of new practices. The India case provided details of the risks of the project and described the barriers in the areas of investment, technology, and practice. Instead of describing the technological and economic barriers, the Indonesia case stated that the project developer would only continue to supporting the project activities for 15–20 years if the CDM project yielded profits. In Mozambique, the rice paddy water management project will be satisfied with one additional aspect, but specific barriers were not provided. In India, project risks in rural areas were considered, and they are described in detail in

the additionality section. This difference in consideration of additionality suggests that the India case was the only project registered under the CDM.

The barriers referred to in Section 4.3.1 refer to those identified in the India case.

### 4.3.1. Investment Barrier

Costs were not included in the case studies. The costs associated with the installation and monitoring of irrigation and drainage facilities, such as water management and educational activities, were higher than revenues, such as CER sales, and acted as a great economic risk to investors who want to promote paddy water management CDM. Owing to the economic burden, the prevalence of new technology in India is typically low as described in the Project Design Document (PDD). The project installed irrigation and drainage infrastructure for water management free of charge without burdening the farmers and provided them with information and training to increase their awareness and empower them. In addition, the project has economic additionality as it generates no financial returns other than profits from CER sales.

### 4.3.2. Technological Barrier

Technological barriers refer to the absence of water management technology. There was lack of technical knowledge required to efficiently use the water needed in paddy fields. The technology for irrigation and drainage infrastructure and monitoring and control devices to systematically manage the water supply were not applied. Barriers to the introduction of new technologies existed as there was a greater awareness of the advantages of using traditional flooded paddy cultivation methods rather than new technologies.

### 4.3.3. Barrier Owing to Prevailing Practices

This barrier is associated with the resistance in implementing new water management practices. The low cost of irrigation water creates no incentive for farmers to conserve water. Adhering to the practice of flooding paddy fields is a traditional cultivation method, and farmers tend to continue and adhere to traditional practices. New technologies undermine confidence in the strengths and knowledge of traditional methods, hence the lack of willingness to take risks associated with innovative practices. Additionally, there was a negative perception that the seed would be washed away by rainwater when temporarily draining the water from the paddy field, and that more weeding work will be required. Project education set to overcome these barriers.

Additionally, we noted institutional absence. There were no government policy regulations or incentive mechanisms to reduce methane through water management. In the absence of essential policies and incentives to implement water management projects, it is impossible to diffuse new technologies. In addition, if the existing project was conducted only as a pilot project funded by government, it could have received additionality in progressing to the CDM project.

## 5. Discussion

The findings of this study highlight a unique characteristic of AWD CDM projects, as they are predominantly implemented in rural areas. Consequently, it becomes crucial to not only manage CDM-related risks, but also address the specific challenges associated with water management projects in these rural contexts. This section focuses on identifying the specific risks involved in an AWD CDM project, shedding light on the potential obstacles and vulnerabilities that must be addressed for effective risk management.

### 5.1. Managing Financial and Technical Risks Stemming from Social and Pollical Factors in Rural Water Management Projects

AWD CDM projects involve economic risks when the operator bears the full cost of transferring the technology required for rice water management to farmers. Owing to the shortage of resources and poverty in rural areas [32,54], the installation of irrigation and

drainage infrastructure is provided free of charge to promote the projects. In addition, the possibility of high maintenance costs, such as electricity required for the operation of irrigation and drainage facilities, increases the economic risk [32,35]. To overcome this, promoting a convergence activity with self-sufficient energy technology such as micro-grids can alleviate economic risks and provide additional economic benefits through the local electricity supply business.

Poor construction methods, contractor performance, and machine operation pose risks in the implementation of technology transfers when conducting international projects such as CDM [32,35]. To mitigate this risk, it is necessary to consider the need to strengthen the capacity of training companies that implement technology in rural areas identified for technology transfer and to carry out technology transfer directly in the field if no local enterprise is available.

The implementation of rice paddy water management projects in rural areas in developing countries poses a potential risk for policy changes, as it is intertwined with other rural issues, such as food production policy, water resource management, and the improvement of rural environmental problems [33,41,42].

Farmer participation is paramount for implementing the monitoring methods outlined in the existing CDM methodology. However, high social risk in rural settings can substantially increase the uncertainty of the results. To mitigate this risk, there is a need for methods that automate monitoring, for instance, through the installation of automated water discharge system, the introduction of technology capable of autonomously monitoring water supply and discharge devices, or the implementation of remote systems allowing data to be automatically collected via satellite communication.

In previous studies, the risk of CDM projects focused on the applicability of the methodology depending on the applied technology [62], and economic risk was treated as the most important factor [63,66]. The results of this study showed that aside from economic risks, barriers to the application of technology in the field pose critical risks as well. The lack of knowledge and implementation experiences in rural areas increases barriers to practice [34,56]. Additionally, declining and aging rural populations further reduce the likelihood of a natural influx of new experiences and knowledge [55]. To overcome this, the risk of difficulties in technology transfer owing to barriers to practice can be alleviated by improving awareness and introducing new knowledge through education and training in rural project implementation (Table 6).

**Table 6.** Risk for AWD CDM projects.

| Category | Risk for AWD CDM Projects |
| --- | --- |
| Financial | Bearing the full cost of technology transfer<br>Providing irrigation and drainage infrastructure<br>Increased maintenance costs, such as electricity required for the operation of irrigation and drainage facilities<br>Economic costs that cannot be compensated by carbon credits owing to fragmented projects |
| Technical | Issues related to construction methods, contractor performance, and machine operation in the implementation of technology transfers<br>Need to strengthen the capacity of training companies that implement technology in rural areas<br>Need to carry out technology transfer directly in the field if no local enterprise is available<br>Difficulties in Farmer Participation Method<br>Eliminating uncertainty in monitoring through agricultural diary |

**Table 6.** *Cont.*

| Category | Risk for AWD CDM Projects |
| --- | --- |
| Social | Barriers to applying technology in the field owing to lack of knowledge and implementation experiences in rural areas<br>Decreased natural influx of new experiences and knowledge owing to declining and aging rural populations<br>Promoting the dissemination and uptake of new water management methods by providing economic incentives to farmers |
| Environmental | Discharge problems of eutrophic substances when discharging water<br>Water supply management |
| Political and institutional | Policy changes related to food production policy, water resource management, and improvement of rural environmental issues |

*5.2. Implications for Global Methane Reduction*

To effectively reduce global methane gas through the AWD CDM project, it is necessary to promote it as a large-scale policy project. The amount of methane reductions that can be achieved, compared to the risk of participating in a fragmented project, will not create merit for the project. A scaled-up approach is required to diffuse CDM projects that reduce methane through coordinated water management in paddy fields. This calls for the establishment of governance for the systematic management of agricultural water resources in large-scale areas that goes beyond the project activities carried out by farmers in paddy field units. It also requires the organic cooperation of multilayered government agencies to formulate and implement policies. Key considerations for scaling AWD technology in large-scale irrigation systems include farmers' willingness to adopt the technology and the existence of favorable operational and environmental conditions [72]. Challenges arise owing to the knowledge-intensive nature of AWD adoption, the necessity for a reliable water supply, the risk aversion of farmers accustomed to traditional practices, and the limitations of pump-based irrigation systems. Neglecting these factors could hinder the widespread adoption of AWD in large-scale irrigation systems, thereby emphasizing the importance of a comprehensive approach to institutional and infrastructure enhancement.

In addition to climate mitigation, methane reduction is closely linked to addressing water scarcity and improving water management governance nationwide. This involves addressing non-point pollution and managing water supplies effectively. Addressing water scarcity necessitates the establishment and improvement of rural water management governance. In rural areas, providing economic incentives to farmers can substantially contribute to the adoption of innovative water management methods and, eventually, higher emission reductions. Strong leadership is paramount to establish multilateral and multi-centric governance at both international and domestic levels.

Recognizing methane reduction as a shared responsibility in the international community is crucial, and concerted efforts are required to curb emissions. This entails implementing global governance mechanisms to execute projects aimed at reducing methane from rice paddy fields in developing countries through technology transfer and financial aid. Specifically, given that rice cultivation CDM projects are considered high-risk projects, garnering private sector participation in the early stages is difficult. High investment risks have been one of the critical reasons for the poor performance of paddy water management projects under the CDM scheme which sought to reduce GHG through market mechanisms. Global society is attempting a goal-based approach such as the GMP for effective methane reduction [5]. The governance approach through the GMP is expected to prompt developing countries to include methane reduction activities via paddy water management in their national methane reduction portfolio. This, in turn, could facilitate the launch of large-scale paddy water management projects. Additionally, de-risking efforts by public sector in the early stages of projects can alleviate the obstacles encountered by the private sector when participating in the projects.

In terms of applicable methodology, Verra, a globally renowned non-profit organization specializing in voluntary carbon crediting, recently announced the permanent inactivation of CDM's rice cultivation methodology, AMS-III.A. This decision was made because of the lack of guidance for land-use stratification, insufficient qualitative information on nitrous oxide emission and carbon stocks in soil, and the absence of a standard guide for methane measurements [73]. These deficiencies could potentially lead to inaccurate and unreliable assessments of methane reductions, thus casting doubt on the overall effectiveness of the methodology. Project developers and private sector investors now have to explore other carbon markets or programs that contribute to recognize the CDM methodology or consider adopting Verra's forthcoming rice-specific methodology when it becomes available.

## 6. Conclusions

This study focused on risks in terms of social and environmental sustainability, which are becoming increasingly important when shifting to the Sustainable Development Mechanism (SDM) system. It builds upon the existing studies that address the risks of CDM projects in the technological and financial sectors. Owing to the policy risks of the country where the project is being conducted, we focused on project risks in the process of registering the project and obtaining CER credits.

In the agricultural sector, the importance of methane reduction projects through water management in rice paddy fields relates not only to GHG, but also to climate change adaptation through water resource management in rural areas. Despite its co-benefits and impact on the contribution to climate change, CDM projects are uncommon, owing to the high risks involved.

The results of this study indicated that beyond economic risks, technological application barriers in the field also pose substantial risks. The lack of knowledge and implementation experiences in rural areas exacerbates these barriers. The risk associated with difficulties in technology transfer owing to barriers can be mitigated by enhancing awareness and introducing new knowledge through education and training in rural project implementation. Moreover, we emphasize the importance of international efforts to establish governance between the private and public sectors and facilitate technology transfers through multi-stakeholder engagement.

This study addressed risk management for rice paddy water management CDM projects in the agricultural sector. This sector has significant potential for global methane reduction but exhibits low implementation performance. To date, only three countries have been considered for demonstration cases conducted as CDM projects. This study is constrained by its data, accessible only via online CDM PDD, and the absence of empirical evidence. These limitations can be addressed in the future as more actual projects are implemented, thereby allowing quantitative analysis that will generate empirical evidence. Despite these limitations, this study holds substantial value as it addressed risk management through qualitative case analysis based on a theoretical approach in the preliminary stages of research.

This study provides specific information to encourage methane reduction worldwide and vitalize rice paddy water management in carbon reduction projects.

**Author Contributions:** Conceptualization, E.-K.J.; methodology, E.-K.J.; validation, E.-K.J.; formal analysis, E.-K.J., E.M.L. and J.K.; investigation, E.-K.J.; data curation, E.-K.J., E.M.L., G.C. and J.M.; writing—original draft preparation, E.-K.J.; writing—review and editing, E.-K.J., E.M.L., J.K., and M.-J.K.; visualization, E.-K.J.; supervision, E.-K.J.; project administration, G.C. All authors have read and agreed to the published version of the manuscript.

**Funding:** This research was funded by National Institute of Green Technology, grant project number C2320301.

**Conflicts of Interest:** The authors declare no conflict of interest.

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
