# Peer review of "Risk Management of Methane Reduction Clean Development Mechanism Projects in Rice Paddy Fields"

_agronomy, doi:10.3390/agronomy13061639_

Round 1

Reviewer 1 Report

Thank you for the opportunity to read this interesting article. The topic of the article seems to be very relevant. The authors have studied the state of this issue in sufficient depth, the list of references contains a sufficient number of articles published over the past 10 years.

It is not often that I have the opportunity to read an article that combines a modern approach to analysis and understanding of the practice of real activity in the field of agriculture. The authors managed to do this. The authors analyzed modern methods of reducing methane emissions in rice fields, possible risks and the attitude of farmers to the proposed measures.

The data given in the article are beyond doubt, the conclusions are quite convincing. The logic of the article is clear and clear.

Author Response

Dear Reviewer,

I appreciate your valuable time and effort in reviewing our manuscript. Your positive feedback is encouraging and means a great deal to us.

We are gratified to hear that you found our topic relevant and the depth of our investigation satisfying. We aimed to strike a balance between a modern analytical approach and an understanding of real-world agricultural practices, and your comments indicate that we were successful in doing so.

We are pleased that you found our data and conclusions convincing, and that the overall structure and logic of our paper were clear. We strived to ensure that our work was comprehensive and accessible to readers, and it is reassuring to hear that we have met these goals.

Your recognition strengthens our belief in the importance of this research and motivates us to continue in our efforts to shed light on critical issues in this field.

Once again, we express our gratitude for your positive feedback and look forward to further opportunities to contribute to this academic discourse.

Sincerely,

Reviewer 2 Report

The subjected paper aim to identify and categorize the risks and barriers to paddy water management in CDM projects in India, Indonesia, and Mozambique. The Authors discuss clean development mechanism (CDM) project (a market mechanism) to reduce methane through private participation and farmers role in the paddy water management projects in rural areas. The topic is very up to date and surely of interest in terms of GHGs emissions and climat changes. It is within the scope of Agronomy Journal. However, I suggest major revision due to some substantial lacks in the presented data and the need of Englih revision in some parts of the paper. In the introduction part some general data concerning the situation of CH4 emissions in the discussed countries is missing.  Authors review the legal situation upon the topic, but it is very general and not related to the particular country situation.E.g. the scale of paddy rice fields agricultural sector emissions would give some background and statistical view of the problem. Later in the discussion part where potential risks are evaluated there are also very general statements. The same in concluding paragraph. The discussion and identified risks are fine but please relate it more to and highlight the situation in the countries under study.

In general the paper needs English revision, especially in terms of repetitions e.g. in Introduction section: methane, collectively, concentrations, antrophogenic etc.

Author Response

Dear Reviewer,

Firstly, we wish to express our gratitude for your insightful and meticulous review of our manuscript. Despite the tight timeframe, we have endeavored to enhance the quality of the manuscript by incorporating your valuable suggestions.

Now, we wish to address the modifications we have made in response to the feedback received during the review.

Reviewer's Comment 1: "In the introduction part some general data concerning the situation of CH4 emissions in the discussed countries is missing. Authors review the legal situation upon the topic, but it is very general and not related to the particular country situation. E.g., the scale of paddy rice fields agricultural sector emissions would give some background and statistical view of the problem."

Response: In the revised introduction, we have included data concerning CH4 emission situations in India, Indonesia, and Mozambique, our case study countries. Further, we have expanded our discussion on the scale of emissions from the non-agricultural sector, providing a statistical overview and underlining the relevance and background of our case study countries.

Reviewer's Comment 2: "Later in the discussion part where potential risks are evaluated there are also very general statements. The same in the concluding paragraph. The discussion and identified risks are fine but please relate it more to and highlight the situation in the countries under study."

Response: We have structured our discussion section more systematically to provide a detailed evaluation of potential risks. We have undertaken a case study analysis across all countries that have applied to the CDM in our research. Our aim was to highlight the risks that need to be mitigated to facilitate global methane reduction and to provide implications from this analysis. We acknowledge that our study still contains general statements. However, as our study is focused on project-level CDM businesses, there were constraints in broadening the context to a national level and relating it to specific national situations. We have addressed this limitation in our concluding section, elucidating the limitations of our data.

Once again, thank you for your constructive feedback. We believe that the manuscript has been significantly improved as a result of these revisions.

Yours sincerely,

Round 2

Reviewer 2 Report

Manuscript has been significalntly improved and hence I recommend its publication in the present form. Congratulations!

The manuscript text contains some minor editing issues, but I believe they will be corrected during proofreading.

E.g. Line 436 Farmer participation of farmers...